# Black Hole Mergers in Holographic Space-time (HST) Models of Inflation

Anish Suresh[1] and Thomas Banks[2]⋆

**1** Rutgers University, New Brunswick
**2** Rutgers University, New Brunswick, New High Energy Theory Center
⋆ tibanks@ucsc.edu

June 10, 2023

## Abstract

We perform a crude computer simulation to show that no problematic black holes are formed by mergers in the early matter dominated phase of the HST models of inflation. These are black holes whose decays could have been seen as signals in the CMB. We also conclude that tiny "black hole galaxies" form. Since black hole decay products are mostly massive standard model particles, and perhaps their superpartners, the fate of these proto-galaxies is a complicated dynamical problem.

## 1  Introduction

In the Holographic Space-time (HST) models of inflation([1]; [2]; [3]; [4]; [5]; [6]), individual inflationary horizon volumes are seen as black holes in the backward Milne coordinates of a slow roll

geometry, as the horizon expands after inflation. Most of these black holes decay, igniting the Hot Big Bang at a temperature around $10^{10} - 10^8$ GeV, but a fraction $\sim 10^{-8}$ of them carry the minimal charge under a discrete $Z_N$ gauge symmetry, and survive to be Primordial Black Hole (PBH) Dark matter, with mass of order $M_P$.

Prior to the Hot Big Bang, there is an era of early matter domination and primordial density fluctuations grow to be $o(1)$ before the black holes decay. The purpose of the present paper is to determine whether that early growth of structure could lead to signals that could falsify the model. That is, one might worry, that the neutral black holes, whose initial mass is $\sim 10^6 M_P$ could combine to form black holes that would decay during an era where they would have left an imprint on observational data. There are very strong constraints(7; 8; 9) on black hole number densities in the mass range whose Hawking lifetime is a few orders of magnitude below the current age of the universe. These problematic black holes could rule our model out.

## 2 Code Structure and Parameters

To determine if most black holes decay before they merge, we wrote a program similar to an $N$-Body simulation on Jupyter Notebooks. Every time-step, we check to see if there exist pairs of black holes that are close enough to merge and adjust their parameters accordingly. This is accompanied by keeping track of the expansion of the universe. Since we do not have computational resources to simulate the entire universe, we make the simulation's topology a 3-D torus. Below, we will formalize these notions.

### 2.1 Parameters

The program starts out with $N$ initial black holes with random initial velocities and masses, all structured in a 3D toroidal lattice. Before we proceed, let us explain the velocity, mass, and position parameters in more detail. We will use natural units in which $\hbar = c = G_N = 1$ throughout.

In order to avoid relativistic effects, we randomly generate velocities from a normal distribution with mean 0 and standard deviation $\frac{1}{10}$ (since we are using natural units, this is equivalent to $\frac{c}{10}$). In Python, this can be done with the `numpy` package, using the function `numpy.random.randn`.

The mass distribution is similar: we randomly produce values from a normal distribution, now with mean $10^6$ and standard deviation of order 1. This mean was not chosen arbitrarily; in HST models, the fluctuations of the cosmological horizon scale $\frac{\delta H}{H}$ are identified with the entropy fluctuations of a black hole whose Schwarzschild radius is equal to the horizon size. The gauge invariant scalar fluctuation parameter is related to these mass fluctuations by a factor of $\epsilon^{-1}$, where $\epsilon \equiv \frac{-\dot{H}}{H^2}$ is the slow roll factor. One argues that the most likely initial conditions have $\epsilon \sim 0.1$. Note that, because of the different power of $\epsilon$ in the scalar to tensor ratio in these models, $\epsilon \sim 0.1$ is compatible with CMB data. The mass fluctuations are also calculated with the aforementioned `numpy.random.randn` function.

The initial time of the simulation $t_0$ can be found with these values. The formula for mass fluctuations is

$$\frac{\delta m}{m} = 0.1 m^{-1} t^{\frac{2}{3}}. \tag{1}$$

Our simulation starts when the fluctuations are of order 1. Plugging in our values gives us

$$1 = 10^{-7} t_0^{\frac{2}{3}}$$
$$\implies t_0 = 10^{\frac{21}{2}}. \tag{2}$$

Since we evenly distribute our black holes at the start of the simulation, another necessary value we need is the length $L$ of each side of the torus. This can be calculated using $n = C\bar{m}^{-3}t^{-2}$, where $n$ is the number density and $C \in [10^{-3}, 10^{-1}]$ is an arbitrary constant. Since $n = NL^{-3}$, we can rewrite our equation as

$$l = \sqrt[3]{\frac{\bar{m}^3 t_0^2}{C}}, \tag{3}$$

where we have solved for the starting length between each adjacent black hole at $t_0$ with the relation $L = \sqrt[3]{N}l$.

With these variables now specified, we can proceed. We then find the acceleration that each particle feels from the Newton's Law of Gravitation, and we can use the numerical integration technique known as Leapfrog Integration to update the positions and velocities of the particles every time-step. This technique, which can be utilized to solve any differential equation of the form $\ddot{x} = f(x)$, has the following steps (for each time-step):

$$\Delta x = \dot{x}\Delta t + \frac{1}{2}\ddot{x}(\Delta t)^2, \tag{4}$$

$$\Delta\dot{x} = \frac{1}{2}\ddot{x}\Delta t. \tag{5}$$

Our scenario allows us to substitute $v = \dot{x}$ and $a = \ddot{x}$. We then compute the new accelerations that the objects feel in their updated positions. Finally, we make sure to update our velocities again with

$$\Delta v = \frac{1}{2}a\Delta t. \tag{6}$$

For more information about this technique, please refer to (10). With some fundamental quantities established, let us move onto more complex parts of the simulation.

## 2.2 Periodic Boundary Conditions

Since our topology is $\mathbb{T}^3 \equiv S^1 \times S^1 \times S^1$, we have to account for some of its properties. The most obvious consequence of this topology is that if a particle leaves our cube, then the particle instantaneously comes back in through the side diametrically opposite. Such a condition helps ensure homogeneity.

Another such property is how particles feel a force from a *boundary* black hole (i.e. black holes that exist on the boundary of our toroidal lattice) from two directions. For example, if there exists a black hole in the middle of our torus and there exists a black hole on the edge, directly to the left of the former particle, then the black hole in the middle feels a force directly from the left and right.

To simplify our process a bit, we utilize the method of images and create *mirror* black holes that are placed diametrically opposite to boundary black holes (it is important to note that they are two different representations of the same black hole). More specifically, each boundary black hole will have exactly one mirror black hole, with the same mass, velocity, and acceleration in our simulation[1]. An important note about acceleration is that a boundary

---

[1]This is actually not true: each boundary black hole on an edge of our cube has 3 mirrors and a boundary black hole on a vertex has 7 mirrors (namely, the other vertices). However, this level of detail makes our program much more complicated with little benefit, so we will work with our approximation.

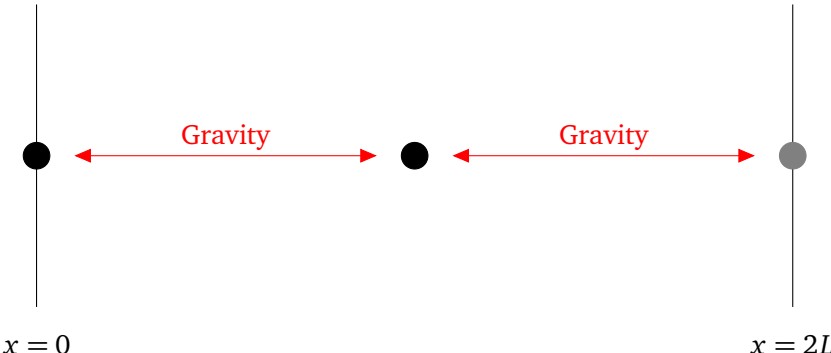

Figure 1: An example to illustrate how forces travel in $\mathbb{T}^1$. There is one boundary black hole on the left and one interior black hole exactly in the middle. Due to the topology, the former can be thought of as being on the right, which we have modelled with a mirror black hole on the right. Forces have periodic boundary conditions as well, so this topology keeps the interior black hole in place, which in turn keeps the stationary black hole in place. This scenario is drastically different than one in $\mathbb{R}^1$, in which the black holes would have gravitated towards each other. Identically, we can think of the force on the right to be from the same black hole, but in the opposite boundary side.

black hole will feel gravity from particles near it and from particles near its mirror (since its mirror is a representation of itself). This detail applied to our earlier scenario can be seen in Figure 1.

Also, this means that out of the $N$ points in our simulation, there exist $\frac{1}{2}[N-(\sqrt[3]{N}-2)^3]$ mirrors. Thus, we are simulating $\frac{1}{2}[N+(\sqrt[3]{N}-2)^3]$ unique black holes. Figure 2 shows what our toroidal lattice looks like at the start when $N = 27$[2].

## 2.3 Merger Process

It is well known that the merging of black holes is a complicated process and highly relativistic. With high accuracy, a simulation of even one pair joining together is extremely detailed. Since we are dealing with a lot of black holes, we will be simplifying this process to one of momentum conservation[3]. The condition to merge is when two black holes get closer than $10m$ (mass of either black hole) of each other. That is, black holes $i$ and $j$ merge when:

$$\text{distance}(i, j) \leq \min(10m_i, 10m_j) \tag{7}$$

The merger then yields a new black hole, with the following parameters:

$$m = m_i + m_j, \qquad x = \frac{1}{2}(x_i + x_j), \qquad v = \frac{m_i v_i + m_j v_j}{m_i + m_j} \tag{8}$$

In addition, if $i$ and $j$ are both boundary and/or mirror black holes, then their mirrored particles, denoted as $i^*$ and $j^*$, will also merge. Obviously, this is a very crude approximation to the actual black hole merger process. We believe that, given our conclusions, it is adequate.

---

[2]The perceptive reader would have noticed the slight issue with Figure 2. Earlier, we assumed that $L = \sqrt[3]{N}l$, which is not $2l$ when $N = 27$. Since we are working with high $N$, we do not have to worry about this discrepancy, since the factor $\sqrt[3]{N}/(\sqrt[3]{N}-1)$ that should have been in (3) is approximately 1 when $N$ is high.

[3]We will not be working with Kerr black holes or binary systems, so we shall only consider conservation of linear momentum.

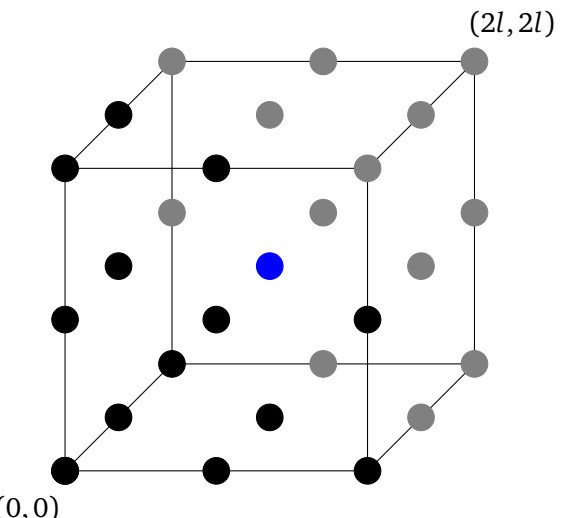

$(2l, 2l)$

$(0, 0)$

Figure 2: The 3D toroidal lattice structure at $t = t_0$ with $N = 27$. Here, we have 13 boundary black holes (in black), which we model on the corresponding diametrically opposite side as 13 mirror black holes (in gray). We also have one interior black hole (in blue).

We will find that the black holes never get close enough to merge, so all the complications of the merger process never come into play.

Before we analyze results, we must consider one other core component of this simulation.

## 2.4 Expansion of the Universe

Since we are simulating the matter dominated era of the universe, the normalized Friedmann's equation tells us that

$$\frac{H^2}{H_0^2} = \Omega_r a^{-4} + \Omega_m a^{-3} + \Omega_k a^{-2} + \Omega_\Lambda \approx \Omega_m a^{-3} \tag{9}$$

$$\implies \dot{a} = H_0 \sqrt{\Omega_m} a^{-\frac{1}{2}} \propto a^{-\frac{1}{2}}. \tag{10}$$

In this simplification, we rightfully assumed that $\Omega_r, \Omega_\Lambda \approx 0$ due to the era of the universe. Similarly, it has been shown that $\Omega_k \approx 0$ in the early universe (11).

Through integration, we arrive at

$$\int a^{\frac{1}{2}} da = \int dt \implies a(t) \propto t^{\frac{2}{3}}. \tag{11}$$

As a result, our expansion is governed by $a = \left(\frac{t}{t_0}\right)^{\frac{2}{3}}$. In the first time-step in which $a \geq 2$, we make $a = 1$ again and double the size of our lattice spacing (which in turn doubles the physical distance between all particles). We allow this process to continue 9 more times; that is, we allow the physical volume of our cube to grow up to $2^{30}$ times the start volume after considerable time has passed. We stop at this point, as the probability of merging after 10 expansions is minuscule.

## 2.5 Time Step

Earlier, we had introduced a $\Delta t$ quantity that refers to the size of each time-step. How do we select an appropriate value for this variable?

This question depends on multiple factors, such as time of the simulation and expansion of the universe. Theoretically, the smaller $\Delta t$ is, the more accurate the simulation is. However, this would mean that the simulation runs for a long time. More specifically, we know that the decay time of these black holes is around:

$$t_d = 2^{10}\pi g^{-1}10^{19} \approx 2^{10}\pi 10^{16},\tag{12}$$

where $g$ represents the number of particle species and is approximately equal to $10^3$. This means that if $\Delta t$ is as 'small' as $10^3$, then the simulation will take $\frac{10^{19}-10^{21/2}}{10^3} \sim 10^{16}$ time steps. If each such time step takes a second, then the program will need more than $3 \times 10^8$ years to finish running. Luckily, we can drastically reduce this value. When running this experiment, we can safely assume that most of the mergers that happen (if any) will happen before the universe expands even once (due to the increase in distance and decrease in acceleration). Thus, a good approximation for the 'end' of our simulation is the time it takes to expand. To find this time, we simply solve for $t$ in this equation:

$$\left(1+\frac{t}{t_0}\right)^{\frac{2}{3}} = 2 \Longrightarrow t \sim 6 \times 10^{10}.\tag{13}$$

Thus, the number of time steps needed is $\frac{10^{10}}{\Delta t}$. We can safely set $\Delta t$ to some value between $10^5 - 10^7$; this is large enough that results are obtainable and small enough that it doesn't negatively impact the simulation[4].

The full code for this simulation is available on GitHub.

# 3 Results

After running the simulation many times with various $\Delta t$ values, there seem to be 0 mergers occurring. To make sure the code was running correctly, we set $C = 10^{19}$ and other high values (which significantly decreases the lattice spacing according to (3)) and noticed that black holes merged together in this setting. With further confirmation of our results, we can now ask: does a complete lack of mergers conceptually make sense?

Let us consider a hypothetical scenario to assess our results. Suppose there are two black hole of mass $10^7$ separated by the distance $l$. Then, note that they merge when the distance between them is less than $10^8$. In addition, suppose that both start out with speed $\frac{1}{4}$ and with velocities such that they are heading towards each other. Finally, since the maximum acceleration they will feel is $\frac{10^7}{10^{16}} = 10^{-9}$, we assume they have a constant acceleration of $10^{-9}$ (again in a way that they are moving towards each other, with increasing speeds). The reason why we are analyzing this scenario is to determine the time taken for two black holes to merge, when they start in the toroidal lattice (given our artificially inflated initial parameters such as velocity, acceleration, mass, etc.).

With $C = 0.1$, we get $l \sim 2 * 10^{13}$, so

$$\frac{2*10^{13}-10^8}{2} = \frac{1}{4}t + \frac{1}{2}10^{-9}t^2 \Longrightarrow t \sim 2 * 10^{11}.\tag{14}$$

---

[4]If two particles are right outside of the merging threshold, then they can move straight past each other without merging when $\Delta t$ is large. This happens due to the high acceleration they obtain from being near each other, which causes a large change in position.

This value is almost double the time it takes to expand the universe, even with unrealistically high parameters. That is, the universe expands before the black holes merge, meaning that the black holes are further apart. These calculations suggest that probability of black holes merging before the universe expands is low.

## 4  Macroscopic Behavior

Up until now, we have been analyzing the behavior of black holes in a tiny portion of the Horizon volume and arrived at the conclusion that black holes do not coalesce in this microscopic setting. However, we are yet to consider the behavior of each 'cube' of black holes; that is, such groups may become bound together, into a structure we will call a 'black hole galaxy.'

This can be determined by comparing two times, vaguely similar to our earlier work. Like before, we will need to calculate the time for the Horizon radius to double. The second time we need is new: the time it takes for mass on the outskirts of the Horizon volume to reach the center via gravitational collapse. For simplicity, we assume that the Horizon volume is spherical. This is then equivalent to a collapse of a Newtonian shell.

### 4.1  Expansion Time

To find the first time, we need an expression for the radius. We know that the coordinate of the horizon grows with the following differential equation:

$$\dot{K} = \frac{1}{a} \tag{15}$$

This can be rewritten as

$$\frac{dt}{da}\frac{dK}{dt} = \frac{dK}{da} = \frac{1}{a\dot{a}} = \sqrt{\frac{3}{8\pi\rho}}a^{-2} \tag{16}$$

by Friedmann's equation. We also know that

$$\dot{\rho} = -3\frac{\dot{a}}{a}\rho \implies \frac{d\rho}{da} = -3\frac{\rho}{a}. \tag{17}$$

With some integration, we end up with

$$\int_{\rho_0}^{\rho}\frac{1}{\rho'}d\rho' = -3\int_{1}^{a}\frac{1}{a'}da' \implies \rho = \frac{\rho_0}{a^3}. \tag{18}$$

(16) now turns into

$$\frac{dK}{da} = \sqrt{\frac{3}{8\pi\rho_0}}a^{-1/2}. \tag{19}$$

This equation is easily solvable, and it gives us

$$K(a) = K_0 + 2\sqrt{\frac{3}{8\pi\rho_0}}a^{1/2}. \tag{20}$$

Now, we can easily obtain the radius $R_H$ of the horizon volume, since the physical distance is simply the coordinate size times the expansion coefficient. That is,

$$R_H(a) = R_0 + 2\sqrt{\frac{3}{8\pi\rho_0}}a^{3/2} = 10m + 2m\sqrt{\frac{3*10^3}{8\pi}}a^{3/2}, \tag{21}$$

which we simplified with the fact that $\rho_0 = \frac{\bar{m}}{(10\bar{m})^3} = 10^{-3}\bar{m}^{-2}$ and $R_0 = 10\bar{m}$, which is the Schwarzschild radius of a black hole with average mass. Our analysis happens when $a \approx 10^5$, meaning that we can safely ignore the first term.

It is easy to see that $R_H$ doubles when $a$ becomes $2^{2/3}10^5$. By plugging in (18) into Friedmann's equation, we get a first order ordinary differential equation. Integrating this gives us

$$\int_0^t \sqrt{\frac{8\pi 10^{-3}\bar{m}^{-2}}{3}}dt = \int_{10^5}^{2^{2/3}10^5} \sqrt{a'}da' \tag{22}$$

$$\implies t_{\text{expansion}} = \sqrt{\frac{3}{8\pi 10^{-3}\bar{m}^{-2}}} * \frac{2}{3}(10^{15/2}) = 2.303 \times 10^{14}. \tag{23}$$

## 4.2 Collapse Time

Now, we need to find the time it would take for mass at the edge of the horizon volume to reach the center. The equation of motion of such a mass is

$$\ddot{x} = -\frac{G(2M)}{x^2}, \tag{24}$$

where $2M = \frac{8}{3}\pi R_H^3 \rho$ is the mass at the center. Luckily, this equation of motion is 1-dimensional and autonomous, so the time it takes for the particle to start at $x = R_H(a = 10^5)$ and end at $x = 0$ can be solved for. This derivation is identical to the one for freefall time, the time it takes for a gas to collapse due to its own gravity. As explained in (12), this has the form

$$t_{\text{ff}} = \sqrt{\frac{3\pi}{32\rho}} = 5.427 \times 10^{14}, \tag{25}$$

when plugged in our values. Notice that $t_{\text{ff}} > t_{\text{expansion}}$[5].

Before we reach a conclusion, we should look more closely at the expression for free fall time. This variable is inversely related to $M$ (so, also $\rho$), and we know this is not fixed. If we recall, $M$ is dependent on $R_H$ and $\rho$, which are both functions of time. Thus, our equation of motion becomes

$$\ddot{x} = -\frac{8\pi G R_H^3(t)\rho(t)}{3x^2}. \tag{26}$$

Now, we get a second-order non-autonomous differential equation, which is much harder to solve by hand, so we can do this numerically. This involves first updating $R_H$ and $\rho$, which will help us update the position $x$.

For the former two, we will be using Runge-Kutta 4, a popular numerical integration technique for first-order differential equations. For example, if we wanted to update $x$ based on its derivative $f(x)$, then we would first create four parameters[6]:

$$k_1 = f(x), \ k_2 = f\left(x + dt\left(\frac{k_1}{2}\right)\right), \ k_3 = f\left(x + dt\left(\frac{k_2}{2}\right)\right), \ k_4 = f(x + dt(k_3)) \tag{27}$$

Now, the updated position is

---

[5]This equation is the general expression for free fall time with mass $M$, but since we have mass $2M$, the 32 should be replaced by a 64. Regardless, the inequality holds true.

[6]These are the parameters if $x$'s derivative is only a function of $x$ and not $t$.

$$x_{\text{new}} = x_{\text{old}} + \frac{1}{6} dt(k_1 + 2k_2 + 2k_3 + k_4). \tag{28}$$

More information about Runge-Kutta 4 can be found here (13).

To effectively use this method, notice that both $R_H$ and $\rho$ can be written as functions of $a$. So, we just have to update $a$, based on its derivative

$$\dot{a} = \sqrt{\frac{8\pi\rho_0}{3}} a^{-1/2}, \tag{29}$$

which of course agrees with (10). With $dt = 10^7$ (which is reasonable, since $dt \ll t_{\text{ff}}$), we get that $t_{\text{collapse}} = 2.989 \times 10^{13}$, as seen in Figure 3 (the code for this calculation can be found on GitHub). With this more accurate model, we find that the time needed for a mass on the boundary of a Newtonian shell is considerably less than our expansion time, meaning that groups of black holes come together and form 'black hole galaxies.'

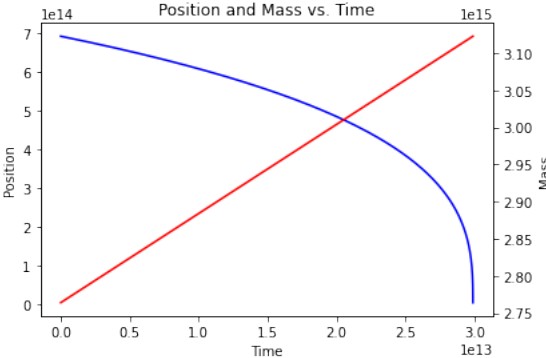

Figure 3: Position $x$ (blue) and Mass $M$ (red) with respect to time. Clearly, $x$ reaches 0 around $t = 3 \times 10^{13}$. Notice that mass increases with time. In fact, it is linear.

This begs the question: why is $t_{\text{collapse}}$ so much smaller than $t_{\text{ff}}$? This can be realized by observing $M$ with respect to time. We already know that:

$$M(a) = \frac{8}{3}\pi R_H^3(a)\rho(a) \propto a^{3/2} \tag{30}$$

Since we know $a$ increases, it is easy to see that $M$ increases as well, which strengthens the force of gravity. Thus, $t_{\text{collapse}}$ is rightfully smaller than $t_{\text{ff}}$. The full evolution of $M$ with respect to time can be seen in Figure 3. Notice the linearity of the red plot. This is in agreement with our work, as (11) plugged into (30) tells us that $M \propto t$.

## 5 Conclusion

The main conclusion of our paper is that the HST model of inflation survives as a model of the early universe. As outlined in previous work(1; 2; 3; 4; 5; 6) it gives us an economical explanation of the CMB fluctuations[7], baryogenesis, and PBH dark matter. The model is fully quantum mechanical, causal and unitary, and has no singularities or Trans-planckian problems. It gives predictions for the detailed form of the tensor fluctuation spectrum and for non-Gaussian fluctuations, which differ from those of field theoretic models. The current paper shows that no problematic black holes, whose decays could have given signals that falsified the model,

---

[7]We do not yet have a precise calculation of the tensor to scalar ratio, but work is in progress on that.

are formed during the early matter dominated era. Intriguingly we have also found that it is likely that bound structures do form during this era. Most of the black holes making up these structures decay, but most of their decay products are actually massive standard model particles, and their superpartners, if those have masses less than $\sim 10^{13}$ GeV. The size of these structures when they are formed is about $10^{-4}$ cm. The structures also contain a sprinkling of Planck mass stable PBHs, Thus, it seems conceivable that these early structures could have an interesting effect on the evolution of structure in the early universe. One would have to understand the rate at which the massive particles cool and whether they remain bound to the PBH clusters, perhaps forming the nuclei around which early galaxies coalesce.

# Acknowledgements

**Funding information**    We thank Profs. A. Brooks and M. Buckley for useful conversations about galaxy formation simulations. This work was supported in part by the U.S. Dept. of Energy under Grant DE-SC0010008.

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
