# Peer review of "Black Hole Mergers in Holographic Space Time Models of Cosmology"

_SciPost Physics Core_

## Round 1 · Referee Report · Anonymous (Referee 1) · 2024-2-23

Strengths

1-Interesting and relevant problem. Necessary to validate a class of inflation model.

Weaknesses

1-Since the article result is highly sensitive to the initial assumptions, these need to be better motivated. Future steps to address paper assumptions require more in-depth discussion.
2-Article presentation, overall tone and grammar should be adjusted to meet journal acceptance criteria.

Report

The holographic model of inflationary cosmology was introduced to explain early structure formation from primordial black holes carrying discrete gauge charge. Too fast black hole merging is a problem for this proposal, since structure would not form fast enough to account for current observation. This article addresses this question through simulation. While the focus on this specific model is noteworthy, the scope of the article remains relatively narrow.

The article presents several insightful ideas and results; however, a primary limitation lies in the dependency of the outcomes to the initial assumptions underlying the study. These assumptions encompass various factors, including the initial velocity, mass, and spatial distribution of black holes, as well as assumptions regarding their non-interaction and the merger process dependent solely on relative distances between black holes. Further justification and exploration of these assumptions are warranted, along with suggestions for future research directions to address these concerns.

Section 3, 4, 5, also needs more details to support the result of the article. For the result in section 3 and 4, it is desirable to perform a sensitivity test for the parameters used in the simulation. The future steps of the article is not explained in the conclusion.

Additionally, the writing style of the article does not meet the publication standards of the journal. Recurring problems are: - The writing are substituted with equations. Equations only should include your symbols up until the end, rather than listing out all steps. Furthermore, please only include only key equations. - General tone too casual. Some more comments regarding grammar, word choices, exposition are provided below.

Considering the content of the article, it is my recommendation that it may not be suitable for publication in Scipost Physics; however, with necessary revisions, submission to Scipost Physics Core could be more appropriate.

Requested changes

1-Abstract - Some words require better substitution (e.g. crude approximation, etc...). - Refrain from using abbreviations (HST, CMB). - " We also conclude that tiny "black hole galaxies form." -> Under what condition? - "..., the fate of these proto-galaxies is a complicated dynamical problem." -> This sentence can elaborate to capture to range covered and not covered in the article.

2-Introduction - Holographic Space-Time needs some explanation. - Be consistent with numbers (10^8 - 10^10 instead of other way around). - For citation, follow journal's standard format. - Refrain from using "our model" when refer to HST.

3-Section 2. - Only include key equations, instead of listing calculations - "Since we do not have computational resources to simulate the entire universe, we make the simulation’s topology a 3-D torus. " -> "The simulation topology is chosen as a 3-D torus to limit the computational resources..." - " One argues that the most likely initial conditions have ε ∼ 0.1. " -> Explain the reasoning here. - "ε ∼ 0.1 is compatible with CMB data" -> include citation - "The mass fluctuations are also calculated with the aforementioned numpy.random.randn function." -> Numpy contains several choices for random number generator. I would include more details here (version, algorithm used, etc.), and reasoning (generation biased important or not, etc.) - Equation 2, the first part can be explained in word rather included in equation. - Equation (4, 5, 6) can be combined - Section 2.3, ", we will be simplifying this process to one of momentum conservation" -> unclear sentence

4-Results need a lot more details. Instead of just a final number, consider the final result sensitivity from input parameters. - This section naming is a bit confusing, may be "Microscopic behavior" is a more accurate name - "After running the simulation many times with various ∆t values, there seem to be 0 mergers occurring" -> Specify the values considered - "we set C = 10^19 and other high values" -> Specific other high values

5-Macroscopic Behavior - same general comments for Results. - The graphs label font need to be consistent with paper font. - ‘black hole galaxy.' -> ‘black hole galaxy'. - "This can be determined by comparing two times, ..." -> "two events/durations" - "The second time we need is new" -> "was not previously considered ..." - ".., we assume that the Horizon volume is spherical. This is then equivalent " -> be more clear on "this" - Equation 15-20 can be combined

6-Conclusion - "The size of these structures when they are formed is about 10−4 cm." - The article used natural unit throughout instead of SI. Comparison/translation at some point in the paper would help reader understand the intuition behind the numbers.

  • validity: good
  • significance: ok
  • originality: ok
  • clarity: low
  • formatting: good
  • grammar: below threshold

---

## Round 1 · Referee Report · Anonymous (Referee 2) · 2024-3-23

Strengths

  1. Novelty of the problem and simulation done.

Weaknesses

  1. Poor writing and presentation. Any new reader interested in the topic will find it very hard to follow.
  2. Very model specific and narrow. Though this is not a negative point but rather just an observation.

Report

There are two competing phenomena, expansion of universe and merger of the blackholes. The author show that blackholes never merge for reasonable initial condition. They also show that there are some sort of configuration
where some blackholes form bound configuration which they call black hole galaxy.

The simulations might have become easier and more intuitive if it was done in co-moving coordinates.

What is being achieved in the paper is rather a very special scenario on a three dimensional Torus. It is not clear if for the full universe, the same conclusions will hold true. Author should justify if anything can be said.

All equations in the paper should be explained properly.

Requested changes

  1. Writing should be improved and more detailed discussion and references should be added.

  2. Every equation should be elaborated and more detailed discussion about the equations should be included.

---

## Round 1 · Referee Report · Anonymous (Referee 3) · 2024-4-1

Strengths

Holographic SpaceTime (HST) is a model by Banks and Fishler (and collaborators) that is a proposed as a potentially UV complete model of cosmology. Since there are no fully accepted UV complete models of cosmology, it is important to check whether this claim is correct.

The present paper addresses one such piece.

Weaknesses

A discussion of HST to set the context is absent.

In my view this is the most important part, because most readers (including the present referee) are unlikely to be sufficiently familiar with HST.

Without an adequate context, it is hard to judge the relevance of the calculation. At the technical level, it becomes difficult to evaluate the credibility of the criteria for choosing the values in section 2.1.

Report

Once the parameters for the black hole gas are chosen (section 2.1), then the calculation is fairly clearly explained, and the code is available to the public on github.

The results (which are numerical) sound reasonable to me, and the authors have discussed their physical plausibility as well.

The eventual fate of "black hole galaxies" (Section 4) will be important to more fully evaluate the status of HST.

Requested changes

I would like to request the authors for a detailed but (semi-)qualitative introduction to HST (say two pages) to be included in the present paper. This will help the reader evaluate this program on their own instead of taking the authors' words on it. This is sufficiently central that in cannot simply be relegated to the references.

In addition, I would like more details of the origin of the choices of the parameters in section 2.1.

Once these two points are addressed, I will be happy to accept the paper -- I am less upset by things like periodic boundary conditions etc (which are standard in QFT etc), since the result seems qualitatively robust (that there isn't any black hole merger).

---

## Round 2 · Referee Report · Anonymous · 2024-6-22

Report

One of the main concern with earlier version was that the paper was not very well written. Current version contains more detailed explanation and also addresses the concerns that were raised .

Recommendation

Publish (easily meets expectations and criteria for this Journal; among top 50%)

---

## Round 2 · Referee Report · Anonymous · 2024-6-28

Report

The paper is much better written now, and even though it does not give as self-contained a review of HST as I had asked for, it can be accepted. It is clear enough, and as I said in my previous report, it has some original ideas.

Recommendation

Publish (easily meets expectations and criteria for this Journal; among top 50%)

---

## Round 2 · Author Response

We have tried to address all of the comments of the referees.

---

## Round 2 · List of Changes

Extensive introduction to the principles of Holographic Space Time Models of Inflation. All units changed to natural units. Numerous other changes.

---

## Editorial Decision

accepted_in_target_journal